# Cellular and Molecular Mechanisms of the Tumor Stroma in Colorectal Cancer: Insights into Disease Progression and Therapeutic Targets

**DOI:** 10.3390/biomedicines11092361

**Published:** 2023-08-23

**Authors:** Nikolay Shakhpazyan, Liudmila Mikhaleva, Arkady Bedzhanyan, Zarina Gioeva, Nikolay Sadykhov, Alexander Mikhalev, Dmitri Atiakshin, Igor Buchwalow, Markus Tiemann, Alexander Orekhov

**Affiliations:** 1Avtsyn Research Institute of Human Morphology, Petrovsky National Research Center of Surgery, 119435 Moscow, Russia; nshakhpazyan@gmail.com (N.S.); mikhalevalm@yandex.ru (L.M.); gioeva_z@mail.ru (Z.G.); drawnman@mail.ru (N.S.); alexandernikolaevichorekhov@gmail.com (A.O.); 2Department of Abdominal Surgery and Oncology II (Coloproctology and Uro-Gynecology), Petrovsky National Research Center of Surgery, 119435 Moscow, Russia; arkady.bedzhanyan@gmail.com; 3Department of Hospital Surgery No. 2, Pirogov Russian National Research Medical University, 117997 Moscow, Russia; rsmu1985@gmail.com; 4Research and Educational Resource Center for Immunophenotyping, Digital Spatial Profiling and Ultrastructural Analysis Innovative Technologies, Peoples’ Friendship University of Russia, 117198 Moscow, Russia; atyakshin-da@rudn.ru; 5Research Institute of Experimental Biology and Medicine, Burdenko Voronezh State Medical University, 394036 Voronezh, Russia; 6Institute for Hematopathology, 22547 Hamburg, Germany; mtiemann@hp-hamburg.de; 7Laboratory of Angiopathology, Institute of General Pathology and Pathophysiology, 125315 Moscow, Russia; 8Institute for Atherosclerosis Research, 121096 Moscow, Russia

**Keywords:** colorectal cancer, conserved oncogenic signatures, gut-associated lymphoid tissue, gut microbiome, metastasis, tumor budding, tumor stroma, Wnt signaling pathway

## Abstract

Colorectal cancer (CRC) is a major health burden worldwide and is the third most common type of cancer. The early detection and diagnosis of CRC is critical to improve patient outcomes. This review explores the intricate interplay between the tumor microenvironment, stromal interactions, and the progression and metastasis of colorectal cancer. The review begins by assessing the gut microbiome’s influence on CRC development, emphasizing its association with gut-associated lymphoid tissue (GALT). The role of the Wnt signaling pathway in CRC tumor stroma is scrutinized, elucidating its impact on disease progression. Tumor budding, its effect on tumor stroma, and the implications for patient prognosis are investigated. The review also identifies conserved oncogenic signatures (COS) within CRC stroma and explores their potential as therapeutic targets. Lastly, the seed and soil hypothesis is employed to contextualize metastasis, accentuating the significance of both tumor cells and the surrounding stroma in metastatic propensity. This review highlights the intricate interdependence between CRC cells and their microenvironment, providing valuable insights into prospective therapeutic approaches targeting tumor–stroma interactions.

## 1. Introduction

Colorectal cancer (CRC) represents a significant global health burden, ranking as the third most common cancer and the second leading cause of cancer-related deaths worldwide. According to the World Health Organization, approximately 1.8 million new cases of CRC were diagnosed, and nearly 900,000 deaths were reported, in 2020 [1]. The high morbidity and mortality associated with CRC can be attributed to several factors, including late-stage diagnosis, limited treatment options, and therapy resistance.

The early detection and diagnosis of CRC are crucial for improving patient outcomes, as the 5-year survival rate for patients diagnosed at an early stage is significantly higher than for those diagnosed at advanced stages [2]. Current screening methods for CRC include fecal occult blood tests (FOBT), fecal immunochemical tests (FIT), flexible sigmoidoscopy, and colonoscopies [3]. While these methods have been effective in reducing CRC incidence and mortality, they are not without limitations. Barriers to CRC screening include patient discomfort, invasiveness, financial constraints, and a low adherence to screening guidelines [4].

Treatment options for CRC primarily depend on the stage of the disease and may include surgery, chemotherapy, radiotherapy, targeted therapy, and immunotherapy. Despite advances in surgical techniques and the development of novel therapeutic agents, the prognosis for patients with advanced or metastatic CRC remains poor, with a 5-year survival rate of less than 20% [5]. Furthermore, the emergence of therapy resistance and the occurrence of tumor recurrence after initial treatment contribute to the challenges associated with CRC management.

Shared characteristics with the stroma of other solid tumors, including cancer-associated fibroblasts (CAFs), tumor-associated macrophages (TAMs), extracellular matrix (ECM) components, and immune cells, mark the tumor stroma in CRC [6]. Despite these commonalities, specific features distinguish the colorectal tumor stroma from other cancer types.

One of the key distinguishing factors is the influence of the microbiome, where bacteria can modulate the tumor stroma by influencing immune cell recruitment and activation, and promote a pro-inflammatory environment [7]. In addition to this, the colorectal mucosa harbors a unique immune system known as the gut-associated lymphoid tissue (GALT) [8]. Disruptions to the balance between immune tolerance and anti-tumor immune responses can lead to alterations in the composition and function of immune cells within the tumor stroma.

A crucial feature that sets CRC apart is the aberrant activation of the Wnt signaling pathway, a hallmark of this disease, especially with mutations in the adenomatous polyposis coli (APC) gene [9]. This pathway can also influence the tumor stroma by promoting the activation of CAFs and the secretion of factors that support cancer cell growth and invasion.

Furthermore, CRC is characterized by tumor budding, identified by the presence of small clusters or single cancer cells at the invasive front of the tumor. This distinctive feature has been associated with a more aggressive phenotype and a worse prognosis [10]. Budding cancer cells can interact with stromal cells, such as CAFs and immune cells, to promote invasion and metastasis [11].

Finally, the CRC stroma displays unique gene expression signatures known as Conserved Oncogenic Signatures (COS). These are specific to the stromal compartment of colorectal tumors and include signatures that reflect an immunosuppressive environment, thereby contributing to the complexity of the tumor’s immune landscape [12,13].

The complexity and heterogeneity of CRC, underscored by the dynamic interplay between cancer cells and the TME, highlight the necessity for a more profound understanding of the disease’s underlying mechanisms. An important aspect, indicative of the tumor stroma’s relevance in CRC evaluation, is the tumor–stroma ratio (TSR). This straightforward marker has emerged as a significant factor in determining CRC prognosis, with a high TSR correlating with an increased risk of cancer recurrence and potential resistance to chemotherapy [14]. The TSR is established by analyzing histological slides, typically from the tumor’s most invasive part, and is categorized into stroma-high (>50% stromal area) and stroma-low (≤50% stromal area) [15]. Recent technological advances, specifically in artificial intelligence, have facilitated automated TSR quantification. This development has proven to be prognostically valid, assisting in clinical decision-making by offering a more objective, standardized analysis, and reducing the workload of pathologists [16].

Nevertheless, the TSR’s exclusive use for patient prognosis remains a matter of de-bate. Various studies suggest that other markers, such as tumor budding, tumor infiltrating pattern, and lymphocyte-to-monocyte ratio—the latter being an independent factor influencing both relapse-free survival and overall survival outcomes—are equally, if not more, critical [17]. Moreover, the reliability of TSR assessment has been questioned due to the poor-to-moderate inter-pathologist agreement [18]. The inclusion of additional markers, such as CAFs or tumor-infiltrating lymphocytes, may result in a more comprehensive patient stratification tool [19,20]. Thus, it is evident that more comprehensive studies are needed to enhance biomarker assessment consistency and validate these findings [18,21]. This intricate scenario involving even a seemingly straightforward and universally accessible marker like the TSR hints at the extreme complexity of the stroma’s cellular and molecular organization when examined at the histological level.

This review will concentrate on these specific features of the colorectal tumor stroma, discussing the cellular and molecular mechanisms that govern its role in CRC progression and therapy resistance. Developing a deeper understanding of these mechanisms may enable the creation of novel diagnostic tools and targeted therapeutic strategies, thereby improving the prognosis for CRC patients.

## 2. The Gut Microbiome and CRC: Dysbiosis, Tumor Stroma Modulation, and Emerging Therapeutic Strategies

The gut microbiome, composed of trillions of commensal microorganisms, plays a vital role in maintaining homeostasis and overall health. Mounting evidence suggests that alterations in the gut microbiome, referred to as dysbiosis, may contribute to the initiation and progression of CRC by modulating the tumor microenvironment (TME), including the tumor stroma [22].

One crucial aspect of this modulation is the direct interaction of specific bacterial species with the tumor stroma. For instance, *Fusobacterium nucleatum*, *Bacteroides fragilis*, and *Escherichia coli* have been closely associated with CRC development, while some species exhibit antitumor activity (Table 1) [23,24,25,26,27]. These bacteria directly interact with stromal cells, including CAFs and immune cells, influencing their activation and function. *F. nucleatum*, for example, adheres to and invades CAFs, leading to the production of pro-inflammatory cytokines like IL-6 and IL-8, which in turn promote cancer cell proliferation, survival, and migration [28,29,30] (Figure 1).

(A): In this illustration, a comprehensive schematic overview of the complex colorectal tumor microenvironment (TME) is provided. The image demonstrates the interaction between bacteria and their metabolites (1 and 2) with the TME, emphasizing the influence of the microbiome on the tumor stroma and inflammatory modulation, including tumor-associated macrophages or TAMs (3). Colorectal cancer (CRC) cells (4) grow at the gut epithelium (5) within the tumor and its stroma. These CRC cells are surrounded by key immune cells, such as TAMs (3) and myeloid-derived suppressor cells (MDSCs) (6). The image also features cancer-associated fibroblasts (CAFs) (7), which support tumor growth and invasion.

The illustration further showcases regulatory T cells (Tregs) (8), which modulate immune responses. Also depicted is the unique mucosal immune system in the colorectal mucosa, represented by the gut-associated lymphoid tissue (GALT) (9) and its interactions with immune cells. Tumor budding, characterized by the presence of small clusters or single cancer cells at the invasive front of the tumor (10), is also portrayed. This visual representation effectively captures the distinctive features and interactions within the colorectal TME.

(B): CRC frequently exhibits driver mutations in Wnt pathway genes, such as APC and β-catenin (CTNNB1). The canonical Wnt/β-catenin pathway is depicted, beginning with extracellular Wnt proteins binding to the cell surface receptors Frizzled (Fz) and the low-density lipoprotein receptor-related protein 5/6 (LRP5/6). Upon Wnt binding, the Fz receptor recruits and activates the intracellular protein Dishevelled (Dvl), leading to the inhibition of the β-catenin destruction complex, which consists of Axin, Adenomatous Polyposis Coli (APC), Glycogen Synthase Kinase-3β (GSK-3β), and Casein Kinase 1α (CK1α). This inhibition prevents the phosphorylation and subsequent degradation of β-catenin, allowing it to accumulate in the cytoplasm and translocate into the nucleus.

Once in the nucleus, β-catenin interacts with T-cell factor/lymphoid enhancer factor (TCF/LEF) transcription factors, activating the transcription of target genes (red asterisk figure in the picture) such as the Transforming growth factor-beta (TGF-β), Connective tissue growth factor (CTGF), and Matrix metalloproteinases (MMPs). These factors influence the tumor microenvironment through the activation of cancer-associated fibroblasts (CAFs), promotion of extracellular matrix (ECM) remodeling, enhancement of cell adhesion, migration, and proliferation, and facilitation of tumor invasion and metastasis.

Another aspect of gut microbiome influence on the CRC stroma is through the production of bacterial metabolites, such as short-chain fatty acids (SCFAs), secondary bile acids, and polyamines [76,77,78,79,80]. SCFAs, like butyrate, exhibit anti-inflammatory and anti-tumorigenic properties by modulating the activation of immune cells and CAFs [76,77,78]. In contrast, secondary bile acids and polyamines promote a pro-inflammatory environment and stimulate reactive oxygen species (ROS) production, leading to DNA damage and the activation of oncogenic pathways in both cancer and stromal cells [79,80,81,82,83].

Dysbiosis can also result in chronic inflammation, a major risk factor for CRC. Pro-inflammatory bacteria stimulate the production of cytokines and chemokines, such as IL-6, IL-8, IL-1b, and TNF-α, which recruit and activate various immune cells, including tumor-associated macrophages (TAMs), T cells, and myeloid-derived suppressor cells (MDSCs) [29,84,85,86]. The complex interplay among bacteria, immune cells, and the tumor stroma creates a self-perpetuating pro-inflammatory and pro-tumorigenic environment, facilitating CRC development and progression.

The gut microbiome can also impact the composition and remodeling of the extracellular matrix (ECM) in the CRC stroma. Bacteria and their metabolites modulate the expression and activity of matrix metalloproteinases (MMPs) and tissue inhibitors of metalloproteinases (TIMPs), which are crucial for ECM remodeling [87,88]. Changes in ECM composition and stiffness influence cancer cell invasion, metastasis, angiogenesis, and immune cell infiltration.

Given the role of the gut microbiome in colorectal tumor stroma development and progression, researchers are actively exploring strategies to manipulate it. Promising approaches include probiotics and prebiotics, fecal microbiota transplantation (FMT), dietary interventions, targeted antimicrobial therapy, and combination therapies.

Probiotics, live microorganisms conferring health benefits when administered in adequate amounts, may help restore immune homeostasis and reduce pro-inflammatory and pro-tumorigenic stimuli contributing to CRC progression [69,89,90]. Prebiotics, non-digestible food components that selectively stimulate the growth and activity of beneficial gut bacteria, promote the production of beneficial bacterial metabolites, such as SCFAs, potentially counteracting dysbiosis’ adverse effects on colorectal tumor stroma development [91,92].

FMT, involving the transfer of fecal material containing a healthy donor’s gut microbiota into a recipient’s gastrointestinal tract, aims to restore the recipient’s gut microbial balance. While FMT has been primarily used for treating recurrent Clostridioides difficile infection, emerging evidence suggests the potential for modulating the gut microbiome in CRC patients, thereby affecting tumor stroma development and disease progression [93,94,95].

Dietary interventions offer another means to influence gut microbial composition and function. Adopting a diet rich in fruits, vegetables, whole grains, and lean proteins, while limiting the intake of processed and red meats, high-fat dairy products, and added sugars, can promote a healthy gut microbiome [96,97,98]. Such dietary changes may potentially reduce inflammation and the risk of CRC by modulating the gut microbiome and its interactions with the tumor stroma.

Targeted antimicrobial therapy, selectively targeting specific pathogenic bacteria implicated in CRC progression such as Fusobacterium nucleatum and Bacteroides fragilis, could be a potential approach to mitigate their influence on the tumor stroma [99,100,101]. However, developing targeted antimicrobial therapies requires a thorough understanding of the complex interactions between these bacteria and the colorectal tumor stroma, as well as the identification of specific molecular targets.

Combination therapies, which involve combining microbiome-targeting interventions with conventional cancer therapies, such as chemotherapy, radiotherapy, or immunotherapy, may enhance treatment efficacy by modulating the tumor stroma and improving the overall tumor microenvironment [102]. These combination strategies could help overcome therapy resistance and improve patient outcomes.

In summary, the gut microbiome plays a crucial role in CRC development and progression by modulating the tumor stroma through direct bacterial interactions, the production of bacterial metabolites, and bacteria-induced inflammation. Dysbiosis can lead to a pro-inflammatory and pro-tumorigenic environment, further promoting CRC. Strategies such as probiotics, prebiotics, fecal microbiota transplantation, dietary interventions, targeted antimicrobial therapy, and combination therapies hold promise for mitigating the gut microbiome’s influence on colorectal tumor stroma development and progression. Further research is needed to optimize these approaches and improve patient outcomes.

## 3. Gut-Associated Lymphoid Tissue (GALT) and Its Influence on Colorectal Tumor Stroma Development and Stability

The gut-associated lymphoid tissue (GALT) is a critical component of the mucosal immune system, responsible for maintaining intestinal homeostasis and protecting the host from pathogens. However, under certain conditions, GALT can contribute to the development and stability of the colorectal tumor stroma, ultimately supporting tumor survival [8].

One way GALT can influence tumor survival is through immune tolerance and immunosuppression. Tumors can exploit GALT’s tolerogenic environment to evade immune surveillance. Regulatory T cells (Tregs) and myeloid-derived suppressor cells (MDSCs) are essential for maintaining immune tolerance within GALT [8,103]. Tumor cells can recruit Tregs and MDSCs to the tumor stroma, where they suppress anti-tumor immune responses by inhibiting the function of cytotoxic T cells and natural killer (NK) cells. This immunosuppressive microenvironment enables the tumor to survive and progress [104].

Additionally, GALT plays a central role in regulating inflammatory responses. In the context of CRC, GALT-driven chronic inflammation can contribute to tumor stroma development and progression. Pro-inflammatory cytokines and chemokines, such as IL-6, IL-8, IL-1β, and TNF-α, can activate stromal cells, including CAFs and tumor-associated macrophages (TAMs). These cells, in turn, support tumor growth, angiogenesis, and metastasis, creating a pro-tumorigenic environment in the colorectal tumor stroma [8,105].

Furthermore, antigen presentation and immune cell infiltration should be considered. GALT contains a high concentration of antigen-presenting cells (AntigenPCs), such as dendritic cells and macrophages, which play a pivotal role in shaping the immune response. In the context of CRC, dysfunctional AntigenPCs may inefficiently present tumor antigens, leading to the suboptimal activation of cytotoxic T cells and a weakened anti-tumor immune response. Additionally, the complex cellular composition of the tumor stroma, including immune cells like TAMs, MDSCs, and Tregs, can impede the infiltration and function of effector T cells, further supporting the tumor immune escape [8,106,107].

Lastly, lymphangiogenesis is a crucial factor. GALT is rich in lymphatic vessels, which are essential for the transport of immune cells and antigens. Tumor cells can exploit the lymphatic network within GALT to facilitate metastasis to regional lymph nodes and distant organs. Moreover, tumor-induced lymphangiogenesis can modulate the tumor stroma, allowing for increased infiltration of immunosuppressive cells and further supporting tumor survival and progression [104].

Given the fundamental mechanisms of GALT’s influence on tumor stroma development, several potential targets within GALT associated with immune regulation and inflammation can be explored as intervention methods. Regulatory T cells (Tregs) and myeloid-derived suppressor cells (MDSCs) play a central role in immune suppression within the tumor microenvironment. Inhibiting their recruitment or function could potentially enhance anti-tumor immunity. Current investigations focus on strategies to deplete or block the function of Tregs and MDSCs, including the use of monoclonal antibodies, small molecules, and immune checkpoint inhibitors [108,109]. The immune checkpoint blockade is a promising strategy for enhancing anti-tumor immunity in various cancers, including CRC. Immune checkpoints are inhibitory pathways that regulate immune responses and maintain self-tolerance. Tumors often exploit these pathways to evade immune surveillance. Blocking immune checkpoints, such as CTLA-4, PD-1, and PD-L1, with monoclonal antibodies has shown promise in multiple cancer types [110].

Another potential target is pro-inflammatory cytokines and chemokines, such as IL-6, IL-8, IL-1β, and TNF-α, which contribute to the promotion of tumor growth, angiogenesis, and metastasis. Targeting these key pro-inflammatory mediators could reduce inflammation within the tumor microenvironment. Inhibitors of these cytokines or their receptors are being explored as potential therapeutic agents for various cancers, including CRC [111]. In addition to inhibition, enhancing the function of AntigenPCs, including dendritic cells and macrophages, could lead to the more efficient activation of cytotoxic T cells and a stronger anti-tumor immune response. Immunotherapeutic approaches, such as dendritic cell vaccines or adoptive cell transfer, aim to improve the antigen-presenting capacity of these cells [112].

Targeting lymphangiogenesis could potentially limit the spread of tumor cells to regional lymph nodes and distant organs and is an attractive approach for treatment. Therapeutic agents targeting the vascular endothelial growth factor (VEGF) family members or their receptors, which play a crucial role in lymphangiogenesis, are being explored as potential treatment options [113,114].

In conclusion, understanding the complex role of GALT in CRC progression offers valuable insights for developing novel therapeutic strategies. By targeting key components of the immune response, inflammation, and lymphangiogenesis within GALT, it may be possible to interfere with the pathological processes that support tumor survival and progression. This approach holds promise for the development of more effective therapies for CRC and may ultimately improve patient outcomes.

## 4. The Role of the Wnt Signaling Pathway in CRC Tumor Stroma Development and Maintenance

The Wnt signaling pathway is a crucial signaling cascade involved in various physiological processes such as embryonic development, cell proliferation, differentiation, and tissue homeostasis. Aberrant activation of the Wnt pathway has been implicated in numerous human cancers, including CRC [115]. In CRC, the Wnt signaling pathway plays a central role in driving tumorigenesis and maintaining the tumor microenvironment. It is also essential for tumor stroma development, remodeling, and stability, making it a critical contributor to tumor progression. The Wnt signaling pathway’s specificity for CRC provides potential molecular targets for regulating this pathway to halt tumor support [116].

The Wnt signaling pathway can be broadly divided into two categories: canonical (β-catenin-dependent) and non-canonical (β-catenin-independent). In CRC, canonical Wnt/β-catenin signaling is the primary pathway involved in tumorigenesis. Under normal physiological conditions, β-catenin levels are tightly regulated through phosphorylation and subsequent degradation. However, in CRC, mutations in genes encoding components of the Wnt pathway, such as APC, CTNNB1 (encoding β-catenin), and AXIN2, lead to the stabilization and accumulation of β-catenin [117]. Consequently, β-catenin translocates to the nucleus, where it interacts with the TCF/LEF family of transcription factors, leading to the activation of target genes involved in cell proliferation, survival, and stemness [118].

The specificity of Wnt signaling in CRC can be attributed to the high frequency of mutations in the APC gene, occurring in approximately 80% of sporadic CRC cases [119]. The APC protein, a key component of the β-catenin destruction complex, plays a crucial role in maintaining intestinal epithelial homeostasis. Loss of APC function due to mutations results in the aberrant activation of Wnt/β-catenin signaling, leading to uncontrolled cell proliferation and the initiation of CRC. Furthermore, the Wnt signaling pathway is intimately involved in the maintenance and regulation of intestinal stem cells, which are essential for tissue regeneration and repair. Dysregulation of Wnt signaling in CRC disrupts the balance between stem cell proliferation and differentiation, contributing to tumor growth and progression [120].

Wnt signaling contributes to the formation and maintenance of the tumor stroma in CRC through various mechanisms. One such mechanism is the crosstalk between tumor cells and stromal cells, where Wnt signaling can mediate communication between tumor cells and stromal cells like CAFs and tumor-associated macrophages (TAMs). In turn, these stromal cells secrete Wnt ligands like Wnt3a, Wnt5a, and Wnt7b, and other factors that further activate Wnt signaling in tumor cells, creating a positive feedback loop that promotes tumor growth and progression [121,122].

Another mechanism involves the regulation of extracellular matrix (ECM) remodeling. Wnt signaling can modulate the expression of matrix metalloproteinases (MMPs) and other ECM remodeling enzymes, which are involved in the degradation and reorganization of the ECM. This remodeling process creates a permissive environment for tumor cell invasion and metastasis [123].

Wnt signaling also plays a role in angiogenesis, promoting the formation of new blood vessels by regulating the expression of pro-angiogenic factors, such as the vascular endothelial growth factor (VEGF) and fibroblast growth factor (FGF) [124,125]. Enhanced angiogenesis within the tumor stroma supports tumor growth by providing nutrients and oxygen while also facilitating the metastatic spread of tumor cells. Additionally, Wnt signaling can influence the tumor immune microenvironment by affecting the recruitment and function of immune cells within the tumor stroma. Activation of Wnt signaling can lead to the recruitment of immunosuppressive cells, such as myeloid-derived suppressor cells (MDSCs) and regulatory T cells (Tregs), which inhibit anti-tumor immune responses and promote tumor progression [126,127].

The critical role of the Wnt signaling pathway in CRC and its significant impact on the tumor stroma has led to the proposition of several molecular targets for therapeutic intervention. These targets aim to inhibit Wnt signaling and disrupt the supportive functions of the tumor stroma, including targets such as Porcupine, Frizzled receptors, Tankyrases, and β-catenin.

Porcupine, an enzyme required for the palmitoylation and secretion of Wnt ligands, serves as one potential target. Inhibitors of Porcupine prevent the secretion of Wnt ligands, thereby obstructing the activation of Wnt signaling. Several such inhibitors, including LGK974 and ETC-159, are presently under evaluation in clinical trials for the treatment of Wnt-driven cancers, including CRC [128,129,130].

Simultaneously, Frizzled receptors, the cell surface receptors that bind to Wnt ligands to activate Wnt signaling, have also been identified as potential targets [129]. Antagonistic antibodies or small molecules targeting Frizzled receptors can prevent Wnt ligand binding and subsequent pathway activation. OMP-18R5 (vantictumab) and OMP-54F28 (ipafricept) represent examples of Frizzled receptor antagonists currently under clinical investigation [131,132].

Moreover, Tankyrases, enzymes that regulate the stability of the scaffold protein AXIN (a component of the β-catenin destruction complex), have been targeted for intervention. Tankyrase inhibitors stabilize AXIN, promoting the degradation of β-catenin and thus inhibiting Wnt signaling. G007-LK and NVP-TNKS656 serve as examples of Tankyrase inhibitors presently in preclinical development [133,134,135].

Lastly, directly targeting β-catenin can impede its interaction with TCF/LEF transcription factors, preventing the activation of Wnt target genes [134]. Several small molecules, such as PRI-724 and BC2059, have been developed to target β-catenin/TCF interaction and are currently being examined in clinical trials [136,137].

In conclusion, the Wnt signaling pathway plays a pivotal role in CRC by influencing the tumor stroma and promoting tumor growth and progression. The specificity of Wnt signaling in CRC, primarily due to the high frequency of APC mutations, makes it an attractive target for therapeutic intervention. Several molecular targets within the Wnt pathway have been identified, and their modulation holds promise for disrupting the tumor-supporting functions of the tumor stroma in CRC. Continued research and clinical trials are needed to fully understand the potential of these therapeutic strategies in treating CRC patients.

## 5. Tumor Budding and Its Influence on the Tumor Stroma in CRC

Tumor budding is a histological feature characterized by small clusters or single cancer cells at the invasive front of the tumor [138]. These budding tumor cells possess features of the epithelial–mesenchymal transition (EMT) and display increased migratory and invasive properties. Tumor budding is considered an important and unique prognostic factor in CRC, as it is associated with aggressive tumor behavior, lymph node metastasis, and poor clinical outcomes. The interaction between tumor budding cells and the tumor stroma is crucial for the progression of CRC [11].

Tumor budding cells exhibit EMT, a process in which epithelial cells lose their polarity and cell–cell adhesion properties and acquire mesenchymal features. This process enhances their migratory and invasive capabilities, ultimately leading to tumor invasion and metastasis. EMT in CRC is driven by various signaling pathways, such as TGF-β, Wnt, and Notch, which are also involved in modulating the tumor stroma [139]. Notably, specific genes, including TP53 and Bcl-2, are recognized as key influencers of tumor budding. TP53 mutations and p53 overexpression in CRC have strong associations with tumor budding, which links them to lymph node metastases and a clinical prognosis [140,141]. These genes influence the process of EMT, thereby affecting tumor invasiveness [142,143,144]. Interestingly, Bcl-2, an anti-apoptotic gene active in stem cells, is implicated in early tumor processes like budding, and its shifting expression during CRC progression underscores its contribution to tumor budding [145]. The crucial roles of both p53 and Bcl-2 in tumor budding and metastasis in CRC highlight their potential as prognostic and therapeutic markers [140,141,142,143,144,145,146,147].

The tumor stroma, composed of CAFs, immune cells, and the extracellular matrix (ECM), plays a crucial role in supporting tumor growth, angiogenesis, and metastasis. Tumor budding cells secrete various growth factors, cytokines, and chemokines that influence the stroma and create a tumor-promoting microenvironment, such as TGF-β, VEGF, IL-6, IL-8, Chemokine (C-X-C motif) ligand 12 (CXCL12) and its receptor C-X-C chemokine receptor type 4 (CXCR4), and Matrix Metalloproteinases (MMPs). In turn, the stromal cells produce factors that enhance the invasive and metastatic potential of the budding cells, such as Fibroblast Growth Factors (FGFs), Hepatocyte Growth Factor (HGF), Platelet-Derived Growth Factor B (PDGFB), Stromal Cell-Derived Factor-1 (SDF-1, also known as CXCL12), Matrix Metalloproteinases (MMPs), Tenascin-C, and Lysyl Oxidase (LOX). This establishes a reciprocal feedback loop [148,149,150,151]. Remarkably, TP53 mutations, which are present in 55–60% of non-hypermutated colorectal cancers, are also implicated in this intricate network [152]. TP53 mutations suppress the transcriptional activity of the wild-type p53 tumor suppressor and have been associated with advanced disease stages and a poor prognosis. These mutations, which are predominantly a missense type [153], lead to the expression of a full-length protein with a single amino acid change and can notably decrease RNA expression levels and alter the tumor microenvironment and immune cell distribution [154,155]. The mutant p53 influences the tumor stroma via various mechanisms. For example, it can interact with other transcription factors such as NF-Y to enhance signaling pathways like JNK/c-JUN, SRC/FAK, and SRC/ERK, thereby promoting stroma remodeling, cancer cell proliferation, and EMT [156,157]. Furthermore, mutant p53 can shape the transcriptional landscape of not only tumor cells but also stroma cells, through the control of genome-wide gene expression via chromatin compaction and interaction with chromatin remodeling complexes [158,159]. Additionally, the secretion of exosomes, particularly miR-1246-enriched exosomes, is increased by mutant p53, contributing to the reprogramming of stroma macrophages to a more cancer-promoting state, favoring immunosuppression [160,161].

Tumor budding is strongly associated with tumor invasion and metastasis in CRC. Budding cells invade the stroma and degrade the ECM through the secretion of MMPs. They can also enter the lymphatic or blood vessels, leading to lymph node and distant organ metastasis. The stromal cells, particularly CAFs, support this process by secreting factors that promote invasion and angiogenesis [148,162].

As is clear from the above, MMTs play a crucial role in the invasive front of the tumor during budding. MMPs, a family of more than 20 distinct enzymes, possess unique properties such as variable substrate specificity, differing expression patterns, activation mechanisms, localization, biological roles, and levels of regulation. These differences grant individual MMPs the ability to influence various physiological processes, including embryonic development, tissue remodeling, wound healing, and immunity. Conversely, pathological roles have also been associated with certain MMPs, including cancer progression, arthritis, and cardiovascular disease [162].

With respect to substrate specificity, MMPs can degrade various types of extracellular matrix (ECM) proteins, albeit with differing efficacies. The differential expression of MMPs, based on cell types and stimuli, furthers the functional diversity of these enzymes. MMPs are initially synthesized as inactive proenzymes (proMMPs), and different MMPs can be activated through distinct mechanisms. Some MMPs operate outside the cell following secretion, whereas others, particularly membrane-type MMPs (MT-MMPs), function at the cell surface following anchoring to the membrane [162,163].

Regulation of MMP activity is intricately controlled at multiple levels, including gene expression, proenzyme activation, interaction with tissue inhibitors of metalloproteinases (TIMPs), and enzymatic clearance from tissues. This functional diversity allows MMPs to impact virtually all stages of colorectal cancer (CRC) progression.

For instance, during tumor initiation, studies have indicated a role for MMP-3 (stromelysin-1), an enzyme capable of degrading various ECM components, including collagen (types II, III, IV, IX, and X), proteoglycans, fibronectin, laminin, and elastin [163]. Overexpression of MMP3, detected in adenomas and early-stage CRC, is suggestive of its involvement in tumor initiation. The implicated mechanisms may include the promotion of inflammation [164,165] or the induction of the epithelial–mesenchymal transition (EMT) [166]. Importantly, MMP-3 efficiently activates proMMP-9, a key player in subsequent stages of tumor progression.

The growth, invasion, and metastasis of tumors may be facilitated by MMPs, notably MMP-2 and MMP-9 (gelatinases A and B), via the promotion of angiogenesis—the formation of new blood vessels crucial for delivering nutrients to growing tumors [125,167,168]. MMP-2 and MMP-9 can degrade ECM proteins, thereby overcoming the physical barriers preventing cancer cell invasion into nearby tissues or metastasis to distant sites. Overexpression of these MMPs has been associated with the invasive and metastatic behavior of CRC, considering their ability to degrade type IV collagen, a major constituent of the basement membrane [169,170]. Moreover, the membrane-bound MMP14 (MT1-MMP) has been implicated in CRC invasion and metastasis.

In addition to promoting angiogenesis and invasion, MMPs may also contribute to immune evasion, a hallmark of cancer, by inactivating cytokines and chemokines, key molecules in immune responses [171,172]. MMP2 and MMP9, in particular, have been linked with immune evasion in CRC.

Furthermore, MMPs, including MMP9 and MMP14, may confer resistance to various cancer therapies by altering tumor vasculature to impact drug delivery, promoting survival pathways in cancer cells, or driving the acquisition of more aggressive phenotypes through EMT [173,174,175]. The specific MMPs implicated in therapy resistance may depend on the type of treatment administered.

In the evolving landscape of therapeutic strategies for colorectal cancer (CRC), the concept of targeting tumor budding emerges as a potential game-changer. This innovative approach seeks to disrupt the tumor–stroma crosstalk, inhibit invasion and metastasis, and ultimately enhance clinical outcomes. Multiple cellular and molecular targets are currently under investigation for their potential to inhibit tumor budding, a phenomenon that plays a critical role in cancer progression.

One such investigative focus involves Epithelial–Mesenchymal Transition (EMT) inhibitors. By interrupting the EMT process, these inhibitors could potentially suppress tumor budding and obstruct invasion and metastasis. Blocking EMT-driving pathways such as TGF-β, Wnt, and Notch can restore the epithelial properties of cells, thus reducing their invasive potential [176,177,178]. A diverse range of therapeutic agents, including small molecules and monoclonal antibodies, are being probed for their potential in targeting these pathways.

Concurrent with these efforts, attention is also centered on cancer-associated fibroblasts (CAFs). These cells play a crucial role in supporting tumor budding by secreting factors that foster EMT, invasion, and angiogenesis. By inhibiting CAF activation, proliferation, or function, it is possible that we might disrupt the tumor–stroma crosstalk and lessen the supportive role of CAFs in tumor budding. Currently, strategies are being formulated to target CAF-derived factors, such as the fibroblast activation protein (FAP) [179] and transforming growth factor-beta (TGF-β) [180].

Matrix Metalloproteinases (MMPs) present another potential target. These enzymes degrade the extracellular matrix (ECM), thereby facilitating tumor invasion and metastasis [181]. Given that budding tumor cells produce MMPs to invade the stroma and promote tumor progression, the inhibition of MMP activity could potentially prevent the degradation of the ECM and reduce tumor budding-associated invasion and metastasis. To that end, several MMP inhibitors have been developed, with some undergoing preclinical and clinical testing for their potential efficacy in CRC treatment [182,183].

The role of immune cells within the tumor stroma, such as tumor-associated macrophages (TAMs) and myeloid-derived suppressor cells (MDSCs), is another area of focus. These cells contribute to the supportive tumor microenvironment and facilitate tumor budding. By modulating the function of these immune cells or reprogramming them to adopt an anti-tumor phenotype, we may inhibit tumor budding and improve CRC outcomes. Immunotherapies, including immune checkpoint inhibitors and adoptive cell transfer, are being studied for their potential to target these immune cells [184,185].

A comprehensive understanding of the tumor budding phenomenon also necessitates an examination of the reciprocal feedback loop between budding cells and the stroma, a process crucial for CRC progression. Disrupting the signaling pathways involved in this crosstalk, such as the chemokine ligand-receptor axis (e.g., CXCL12-CXCR4), may inhibit tumor budding and its associated invasion and metastasis. Consequently, therapeutic agents that target these signaling pathways are being investigated [186,187].

Lastly, the role of Tumor-Initiating Cells (TICs) or cancer stem cells is being scrutinized. These cells, a subpopulation of tumor cells with self-renewal and differentiation capabilities, drive tumor heterogeneity and resistance to therapy. Tumor budding has been linked to the presence of these TICs, which can initiate new tumor growth at the invasive front. Strategies aimed at targeting TICs using specific surface markers, such as CD133, CD44, and Lgr5, or cellular processes like self-renewal and differentiation, are being explored to inhibit TICs and subsequently, tumor budding [188,189,190]. Targeting metabolic reprogramming in TICs, such as glucose metabolism, glutamine metabolism, or fatty acid synthesis, is another promising approach. By selectively affecting TICs, we might impair their survival and function, furthering our progression towards an effective treatment [191,192].

In conclusion, tumor budding is an essential and unique feature of CRC that significantly influences the tumor stroma and contributes to aggressive tumor behavior, invasion, and metastasis. Several potential cellular and molecular targets are being investigated to stop tumor budding and its supportive role in CRC progression. Targeting tumor budding may disrupt the tumor–stroma crosstalk, inhibit invasion and metastasis, and ultimately improve the clinical outcomes of CRC patients. Further research is required to identify and validate effective therapeutic strategies that specifically target tumor budding and its underlying molecular mechanisms in CRC.

## 6. Conserved Oncogenic Signatures in CRC Stroma

The CRC stroma displays unique gene expression signatures known as Conserved Oncogenic Signatures (COS), which are specific to the stromal compartment of colorectal tumors. These signatures represent consistent and reproducible patterns of gene expression that have been identified across various CRC patients and studies [193,194,195]. The presence of COS suggests that there are specific interactions between CRC cells and their stromal components, which may drive tumor progression and influence treatment outcomes. Key players in this signature include Fibronectin (FN1), Matrix metalloproteinases (MMPs), the Vascular endothelial growth factor (VEGF), the Transforming growth factor-beta (TGF-β), and Interleukins (e.g., IL-6, IL-8) [19,196,197,198].

Although COS represent consistent and reproducible patterns of gene expression, there may still be some degree of inter-patient variability due to factors such as genetic background, lifestyle, and tumor stage. These inter-patient differences in COS may influence treatment response and outcomes, highlighting the importance of personalized medicine approaches. By analyzing the unique molecular characteristics of each patient’s tumor, including their specific COS, clinicians can develop tailored treatment plans that maximize therapeutic efficacy and minimize side effects.

For example, some of the genes identified in the COS are involved in extracellular matrix remodeling, which is known to play a critical role in facilitating tumor invasion and metastasis [199]. These genes include matrix metalloproteinases, which degrade various components of the extracellular matrix; lysyl oxidase, which contributes to collagen cross-linking and stabilization; fibronectin, a major component of the extracellular matrix that mediates cell adhesion and migration; and collagens, the primary structural proteins in the extracellular matrix (Table 2). By targeting the specific genes or pathways involved in this process, it may be possible to inhibit tumor progression and improve treatment outcomes.

Moreover, the study of COS in CRC stroma has revealed the potential role of the immune system in CRC development and progression. Some of the gene expression signatures identified are associated with immune cell infiltration, immune cell activation, and immune checkpoint pathways, indicating that the immune system may play a significant role in shaping the tumor microenvironment (Table 2). Targeting these immune-related genes or pathways may help modulate the immune response and enhance the effectiveness of immunotherapies in CRC patients.

Another important aspect of COS in CRC stroma is the identification of genes and pathways involved in angiogenesis. By identifying specific angiogenesis-related genes in the COS, researchers can better understand the mechanisms driving this process and develop targeted therapies to inhibit angiogenesis in CRC patients (Table 2).

In conclusion, COS in CRC stroma is a distinct concept from the previous chapters on the Gut Microbiome, Gut-Associated Lymphoid Tissue (GALT), Wnt Signaling, and the Tumor Budding. While the previous chapters focus on specific components or processes within the tumor microenvironment, COS represents a unique gene expression signature specific to the stromal compartment of colorectal tumors. Although there may be some overlap or interplay among these topics, COS is an individual mechanism that could potentially serve as a therapeutic target.

While Table 2 presents a meticulously curated selection of key cytokines, growth factors, and other molecules integral to CRC’s biology, it is important to acknowledge that this is not an exhaustive list. The complex landscape of CRC involves a multitude of additional molecules which significantly contribute to its development, progression, and prognosis.

The transforming growth factor-beta (TGF-β), for instance, is a potent cytokine that can suppress tumor development in early stages of CRC, while promoting disease progression in advanced stages. Interleukins such as IL-6 and IL-10 also play significant roles in modulating the tumor microenvironment through the promotion of inflammation or immunosuppression.

Critical regulatory proteins involved in signaling pathways, such as Wnt/β-catenin, Notch, Hedgehog, and PI3K/AKT/mTOR, command a substantial influence on cell proliferation, differentiation, and survival, and their dysregulation is a common feature in CRC.

Moreover, the role of molecules involved in cellular metabolism cannot be overlooked. Proteins such as glucose transporter 1 (GLUT1), lactate dehydrogenase A (LDHA), and monocarboxylate transporter 4 (MCT4) are central to the metabolic reprogramming seen in cancer cells, marking them as significant players in CRC pathophysiology.

Furthermore, the emerging field of epigenetics has identified molecules such as DNA methyltransferases (DNMTs), histone deacetylases (HDACs), and various non-coding RNAs as contributing factors to CRC progression.

Thus, while Table 2 highlights a subset of pivotal molecules in CRC, it is crucial for the readers to appreciate the broader, intricate matrix of factors that constitute the molecular pathogenesis of CRC.

This multifaceted understanding can potentially pave the way for novel therapeutic targets and strategies.

## 7. The Hypoxia Effect on Tumor Stroma in CRC

Hypoxia, a low oxygen condition, is an integral feature of the TME in CRC. This hypoxic state is created by rapid cellular proliferation outpacing the rate of angiogenesis, leading to a mismatch between oxygen supply and demand. The adaptations to hypoxia contribute to an aggressive tumor phenotype and a hostile TME, exacerbating disease progression and therapy resistance [231].

One crucial consequence of hypoxia is a metabolic shift. Tumor cells and stromal cells adapt to oxygen deprivation by switching from oxidative phosphorylation to glycolysis, a phenomenon known as the Warburg effect [232]. In CRC, the hypoxia-inducible factor 1-alpha (HIF-1α) mediates this shift by upregulating genes such as glucose transporter 1 (GLUT1) and lactate dehydrogenase A (LDHA), enhancing glucose uptake and lactate production, respectively [233]. This not only allows for survival in hypoxic conditions but also promotes tumor growth and immune evasion by acidifying the TME [234].

ECM, a non-cellular component of the TME, is also heavily influenced by hypoxia. In CRC, hypoxia can induce CAFs to increase the synthesis of collagen and other ECM proteins, resulting in a stiffer and more fibrotic stroma. This altered ECM structure can enhance the motility of CRC cells and facilitate their metastatic spread [235].

Hypoxia can also induce dormancy in a subset of CRC cells [236], causing them to enter a quiescent state that makes them resistant to therapies targeting rapidly dividing cells. This can contribute to minimal residual disease and tumor recurrence, as these dormant cells can reawaken and repopulate the tumor. In CRC, the transcription factor DEC2 (Differentiated Embryo-Chondrocyte Expressed Gene 2) is known to promote hypoxia-induced dormancy [237].

Epigenetic modifications, another adaptation to hypoxia, can have long-lasting effects on gene expression in CRC. For example, hypoxia can induce the upregulation of DNA methyltransferases, leading to hypermethylation and silencing of tumor suppressor genes [238]. Hypoxia can also influence histone modifications and the expression of non-coding RNAs, altering the expression of genes related to survival, angiogenesis, and metastasis [239,240].

Lastly, hypoxia is spatially heterogeneous within a tumor, with areas of severe hypoxia interspersed with well-oxygenated regions. This heterogeneity can affect the efficacy of anticancer therapies and contribute to the heterogeneous response to therapy observed in CRC [241].

Given the influence of hypoxia on angiogenesis, recent clinical practices have incorporated antiangiogenic strategies. One such therapy is Bevacizumab, an anti-VEGFR therapy. Bevacizumab is a monoclonal antibody that inhibits VEGF, resulting in a decrease in the number of blood vessels supplying the tumor and hence, slowing tumor growth and progression. Its use has been found effective in improving survival rates in metastatic CRC cases when used alongside chemotherapy [242].

Arterial embolization, another strategy, involves blocking the blood vessels supplying the tumor, causing tumor ischemia and cell death. This technique has been effectively employed in managing liver metastases from CRC and is used alongside chemotherapy or ablation [243,244].

Despite the promise of these antiangiogenic approaches, they underline the complexity of targeting tumor angiogenesis. Future research should aim at identifying strategies to enhance the efficacy of these therapies, overcome resistance, and ultimately improve patient outcomes.

## 8. The Role of the TME in Multidrug Resistance in Colorectal Cancer

Multidrug resistance (MDR) is a significant challenge in CRC treatment, often leading to sub-optimal patient outcomes. There is growing recognition that TME, including various cellular components and ECM, substantially influences drug responses.

Traditionally viewed as a mere structural scaffold, the ECM is now known to dynamically regulate cellular behavior. Changes in the composition and stiffness of the ECM, collectively known as ECM remodeling, contribute to drug resistance. Increased ECM stiffness can shield cancer cells from the cytotoxic effects of chemotherapy. Mechanotransduction, the process of converting mechanical stimuli into biochemical signals, could underlie this form of resistance. This process activates several intracellular signaling pathways, enhancing cell survival and contributing to drug resistance. In response to ECM stiffness [245], integrins (cell-ECM interaction mediators) activate the Focal Adhesion Kinase (FAK) Pathway. This activation promotes cell survival and proliferation through downstream signaling molecules like ERK and PI3K/AKT, making cancer cells more resistant to chemotherapy-induced apoptosis [246].

Moreover, ECM stiffness has a direct impact on another important pathway known as the PI3K/AKT Pathway. It can activate this pathway either directly or indirectly through FAK. The PI3K/AKT pathway is well-known for its role in promoting cell survival, proliferation, and growth [247]. As a result, when this pathway gets activated, it enhances the resistance of cancer cells to various chemotherapeutic agents.

Additionally, the Rho/ROCK Pathway plays a significant role in the process of mechanosensing ECM stiffness. It regulates cytoskeletal tension and cell contractility, further contributing to cell survival and chemoresistance [248]. This pathway provides an additional layer of complexity to the cellular response, allowing cancer cells to adapt and resist the effects of chemotherapy.

Furthermore, the Hippo-YAP/TAZ Pathway also comes into play in response to ECM stiffness. Key effectors in this pathway, YAP and TAZ, play vital roles in regulating cell proliferation and apoptosis. Interestingly, ECM stiffness can inhibit the Hippo pathway, leading to the nuclear accumulation of YAP/TAZ and promoting the transcription of survival and chemoresistance genes [249]. This mechanism adds yet another dimension to the intricate network of signals that contribute to enhanced cell survival and drug resistance.

CAFs and MDR. CAFs significantly contribute to the tumor stroma and can alter drug responses. They contribute to a fibrotic and dense tumor stroma, which can physically restrict drug penetration. In CRC, CAFs have been shown to secrete various factors, including ECM proteins and growth factors, that can protect cancer cells from the cytotoxic effects of drugs like 5-Fluorouracil (5-FU) and oxaliplatin [250,251]. Moreover, CAFs can physically limit drug delivery to the tumor by producing a dense, fibrotic stroma, further contributing to resistance. They can also influence the tumor immune microenvironment, secreting factors such as CXCL12, which can limit the penetration of cytotoxic T cells into the tumor, thereby reducing the effectiveness of immunotherapies [252].

Immune Cells and Drug Resistance. The immune components of the stroma can significantly shape the drug response in the tumor microenvironment. Certain immune cells, such as tumor-associated macrophages (TAMs), can secrete factors that protect cancer cells from the cytotoxic effects of chemotherapy [253]. TAMs, often adopting an M2-like phenotype within the tumor microenvironment, have been associated with resistance to anti-angiogenic therapy like bevacizumab in CRC [254]. This resistance might be due to TAMs’ ability to secrete alternative angiogenic factors that can maintain blood vessel formation [255], despite the VEGF pathway being blocked by bevacizumab. Additionally, TAMs can secrete factors like the epidermal growth factor (EGF) [256], which can trigger survival pathways in cancer cells, rendering them resistant to anti-EGFR drugs.

Other immune cells, like regulatory T cells (Tregs), could be involved in resistance to immunotherapies such as checkpoint inhibitors. Tregs can secrete immunosuppressive cytokines like the transforming growth factor-beta (TGF-beta) and IL-10 [257], which can hamper the activity of cytotoxic T cells and natural killer (NK) cells. By stifling these anti-tumor immune responses, Tregs could limit the efficacy of immunotherapies.

Myeloid-Derived Suppressor Cells (MDSCs) represent another key player in immune-mediated drug resistance. MDSCs are a heterogeneous population of cells that expand during cancer and other disease states. These cells are known for their immunosuppressive activities, particularly their ability to inhibit T cell responses [258]. MDSCs can contribute to chemoresistance through various mechanisms. Moreover, MDSCs can promote a pro-tumoral immune environment by secreting factors that inhibit cytotoxic immune responses and support tumor growth, thereby further exacerbating resistance to both chemotherapy and immunotherapy [255].

Stromal Contributions to Drug Efflux. Drug efflux is a well-established mechanism of MDR. Stromal cells within the TME, such as CAFs or TAMs, can contribute to this process by upregulating the expression of drug efflux pumps in cancer cells, thereby decreasing the intracellular concentration of chemotherapeutic agents and promoting resistance. This upregulation often involves the ATP-binding cassette (ABC) transporters family, including P-glycoprotein (P-gp), multidrug resistance protein 1 (MRP1), and breast cancer resistance protein (BCRP) [259,260,261,262], whose overexpression leads to decreased intracellular drug accumulation, thus contributing to the MDR phenotype.

In conclusion, the TME significantly influences MDR in CRC through various mechanisms, involving ECM remodeling, CAFs, immune cells, and drug efflux systems. By targeting these areas, it may be possible to improve the effectiveness of existing therapies and patient outcomes. Each of these avenues offers potential strategies to overcome MDR in CRC, highlighting the need for further research in this critical area.

## 9. Interplay between Tumor Stroma and Immunotherapies in Colorectal Cancer: Implications for Checkpoint Blockades, CAR T-Cells, NK Cells, and CAR Macrophages

The landscape of CRC treatment is rapidly changing with immunotherapy innovations like checkpoint inhibitors and cell-based therapies. However, challenges posed by the tumor microenvironment demand novel strategies. Researchers are now targeting key players within the tumor, exploiting the infiltrative nature of macrophages, and leveraging the potency of Natural Killer cells, painting a hopeful future for CRC treatment.

Immunotherapy has revolutionized the approach to colorectal cancer treatment, significantly advancing therapeutic strategies and introducing novel opportunities for patient outcomes. One field of interest is checkpoint blockade immunotherapy, which has gained substantial attention due to the efficacy of immune checkpoint inhibitors (ICIs) in advanced tumors [263,264]. Pembrolizumab and Nivolumab, PD-1 inhibitors, have exhibited promising results, especially for metastatic colorectal cancer [242]. Despite these achievements, the response to these treatments varies among patients, indicating the need to identify reliable predictors for determining patient suitability and expected benefits from ICIs [264].

Understanding the TME is essential for elucidating the intricacies of colorectal cancer and the application of ICIs. The TME comprises a diverse array of cell types and signaling molecules, significantly influencing the efficacy of ICIs [265]. Central to the TME are cancer-associated fibroblasts (CAFs), derived from various cells such as resident fibroblasts, epithelial cells, and mesenchymal stromal cells [266,267,268,269]. These CAFs contribute substantially to tumor angiogenesis, immune suppression, and drug access, impacting the TME’s dynamics [266,267].

Given the critical role CAFs play in immune suppression, strategies aiming to reprogram these cells have exhibited promise in enhancing T cell activation and infiltration, potentially improving patient prognosis [269]. Moreover, recent developments in single-cell RNA sequencing technologies have paved the way for a more comprehensive understanding of the diverse CAF subsets and their roles in immune modulation [266,269].

A wide range of immune cells, such as T cells, B cells, and natural killer (NK) cells, infiltrate tumors, contributing to enhanced antitumor immunity and improved responses to ICIs [268,270]. The interactions between these immune cells and tumor cells are mediated by numerous chemokines, chemokine receptors, and immune checkpoint molecules. Notably, genes such as CCL11, CCL19, CCL22, CCL28, CXCR5, IDO1, LAG3, and TIM4 form a signature associated with immunotherapy response [271,272]. Heat shock proteins, specifically HSP70 family members like HSPA1A, HSPA8, and HSPA9, are also part of this gene signature [270].

Tumor-associated myeloid cells (TAMCs) are another essential component of the TME, comprising neutrophils, monocytes, basophils, eosinophils, and macrophages [273,274]. Their functions in tumor progression are context-dependent, determined by the TME’s state. Notably, some TAMCs, such as tumor-associated macrophages (TAMs), myeloid-derived suppressor cells (MDSCs), and tumor-associated monocytes, express immune checkpoints like PD-1 and CTLA-4, which can inhibit T cell and NK cell cytotoxicity [273,274].

Furthermore, tumor-associated endothelial cells (TECs) play a significant role in tumor angiogenesis, hampering the immune response by limiting leukocyte adhesion to tumor epithelial layers and reducing the infiltration of effective T cells into the tumor site [275]. Anti-angiogenesis therapy may improve the efficacy of ICIs in tumors that overexpress VEGFA [276]. Moreover, CAFs contribute to fibrotic stroma formation within the TME, leading to increased tumor tissue stiffness, drug resistance, chronic inflammation, and cancer cell proliferation [273,274].

Regulatory T cells (Tregs) are also central players in modulating the immune system [277,278]. A high density of Tregs within a tumor often indicates an unfavorable prognosis, potentially resulting in a low efficacy of ICIs [279,280]. Interestingly, tumor-derived exosomes (TEXs) foster an immunosuppressive TME, facilitating intercellular communication, contributing to angiogenesis, metastasis, and chemo-resistance, thereby undermining the effectiveness of the immune checkpoint blockade [281,282].

Chimeric Antigen Receptor (CAR) T-cell therapy has demonstrated considerable promise in addressing hematological malignancies and is now shifting its focus to solid tumors, such as CRC [283]. The emphasis has traditionally been on directly neutralizing cancer cells, but recent approaches have honed in on the tumor stroma—the non-malignant cellular framework that supports tumor growth and progression. The tumor stroma has been recognized as a ripe target for therapeutic interventions, a development that is echoed in the design of new CAR T-cell constructs.

A novel strategy involves targeting the fibroblast activation protein (FAP), abundantly expressed within the tumor stroma [284]. Using genetically engineered T cells that express FAP-specific CARs, we can instigate an anti-tumor response that diminishes stromal cells and suppresses tumor growth. Intriguingly, these modified T cells are capable of secreting cytokines such as Interferon-gamma (IFN-γ), thereby bolstering endogenous CD8+ T-cell responses and eliciting a more robust immune response against the tumor [284].

In a parallel development, CAR T cells are being tailored to deposit interleukin-12 (IL-12) within tumor sites upon activation [285]. The localized release of this potent cytokine induces an innate immune response towards cancer cells that would otherwise remain undetected by CAR T-cells [286]. This strategy potentially addresses a key limitation of CAR T-cell therapy: the inability to target every cancer cell due to the vast phenotypic diversity within solid tumors [287].

However, there are substantial challenges curtailing the success of CAR T-cell therapy in solid tumors. Factors such as tumor antigen heterogeneity and the suppressive tumor microenvironment pose significant obstacles [288]. The antigenic profile can fluctuate dramatically among different cancer cells within a single tumor and between different regions of the same tumor. The tumor microenvironment also constitutes a hostile territory for T cells, which can become suppressed or “exhausted” due to chronic exposure to tumor-derived factors [288].

Addressing these challenges necessitates combined therapies or multifaceted strategies. For instance, treatment regimens could encompass CAR T-cells designed to target multiple tumor antigens, or combination treatments aimed at both cancer cells and stromal cells like FAP [289].

In another innovative approach, the tumor-homing capabilities of mesenchymal stem cells (MSCs) are harnessed [290]. By genetically modifying MSCs to release immuno-modulatory proteins such as IL7 and IL12, it could be possible to enhance the expansion and activation of CAR T-cells within the tumor. This promising strategy signifies a critical advancement in our effort to bolster the efficacy of CAR T-cells in the treatment of solid malignancies, thus warranting further research and development [290].

Natural killer (NK) cells, integral components of the innate immune system, are being recognized as potential influencers in the fight against CRC [291,292]. This interest is driven by the unique tumor microenvironment in CRC, a composite of stromal cells that simultaneously facilitates tumor growth and impedes immune responses.

In the development of NK cell-based therapies, one notable example is the surface antigen TEM8, which is characteristically expressed within the CRC tumor stroma. A novel strategy involves engineering a Tri-specific Killer Engager (TriKE), termed ‘cam1615TEM8’, that precisely targets TEM8-expressing cells within the tumor milieu. This agent instigates NK cell degranulation and inflammatory cytokine production, thereby potentially enhancing NK cell-mediated cytotoxicity against colorectal tumor spheroids [291].

However, the design of targeted NK cell therapies for CRC presents significant challenges, primarily due to the dearth of specific markers. Consequently, current research is investigating alternative strategies to augment the therapeutic activity of NK cells within the CRC microenvironment. Approaches under consideration include the employment of cytokine-based agents, NK cell-engager molecules, and immune checkpoint inhibitors [292,293,294].

Cytokines like IL-12 and IL-18 have demonstrated effectiveness in amplifying NK cell cytotoxicity against solid tumors, including CRC. Moreover, deploying monoclonal antibodies (mAbs) that specifically target CRC antigens could render these tumors more susceptible to NK cell-mediated lysis [295].

Research also explores immune checkpoint inhibitors aimed at subduing NK cell-inhibitory signals, thereby enhancing NK cell-mediated immune responses [295]. Numerous clinical trials are currently underway, assessing the efficacy of different mAbs against these inhibitory signals in advanced and metastatic CRC cases [295].

Simultaneously, innovative strategies such as the use of bispecific mAbs, NK cell-engaging antibodies, and antibody-cytokine fusion proteins are being examined. These could potentially augment the efficacy of NK cells in identifying and eradicating CRC cells [295].

Parallel investigations are venturing into other aggressive cancers, such as pancreatic ductal adenocarcinoma (PDA), infamous for its invasive and metastatic nature [296]. Research has indicated the role of the multifunctional protein Gas6, chiefly produced by tumor-associated macrophages (TAMs) and cancer-associated fibroblasts (CAFs), in propelling PDA metastasis. Gas6 signaling inhibition appears to confer dual benefits: potentially reversing the epithelial-to-mesenchymal transition (EMT) of tumor cells and promoting NK cell activation [296].

Overall, while the burgeoning field of NK cell therapy in treating CRC and other cancers presents great promise, it also poses significant challenges. As our comprehension of the complex interactions between immune and stromal cells within the tumor microenvironment continues to evolve, the vision of more effective, targeted, and personalized treatment strategies comes into sharper focus [297,298].

The advent of adoptive cell therapy has achieved remarkable successes in hematological malignancies, yet significant strides in treating solid tumors, including CRC, remain to be made [299]. Factors contributing to this disparity include challenges in CAR T cell manufacturing, the lack of tumor-specific antigens, suboptimal CAR T cell infiltration into tumor sites, and the immunosuppressive nature of the tumor microenvironment [299]. However, the recent development of CAR macrophage therapy may signal a promising advancement in the treatment of solid tumors [212,299].

Distinct from T or natural killer cells, macrophages are adept at infiltrating tumors, constituting up to 50% of the tumor microenvironment in cancers such as melanoma, renal, and colorectal cancer [300]. Combined with their ability to incite potent anti-tumor immune responses and remodel the tumor microenvironment, they represent a compelling candidate for CAR engineering [212,300].

The exploration of CAR macrophage therapy for CRC encompasses two main strategies. The first entails transducing the THP-1 human macrophage cell line with an anti-CD19 CAR encoding the CD3ζ intracellular domain [301,302]. Alternatively, the second approach utilizes peripheral blood CD14+ monocytes and introduces CARs via an adenoviral vector [301]. Both methods have been evaluated in in vivo models, where CAR-macrophages have shown significant reductions in the metastatic tumor burden [301].

The heterogeneity of tumor-associated macrophages (TAMs) represents another intriguing area of study. Certain TAM states have been associated with specific clinical outcomes, suggesting potential therapeutic opportunities [303]. Additionally, the interplay between TAMs and other immunosuppressive cells, such as myeloid-derived suppressor cells, in modulating the tumor environment underscores the pivotal role of macrophages in tumor progression and therapy response [304].

Despite these advancements, obstacles persist. Enhancing the migration and longevity of CAR-macrophages within solid tumors remains a formidable challenge. The ATAK platform offers a promising solution in this direction, enabling the development of myeloid cells with innate immune receptor-inspired CARs against cancer cells and primed monocytes to instigate T cell responses [305,306,307,308,309,310,311,312].

Though CAR macrophages, like their T and NK cell counterparts, possess limitations, their unique advantages, particularly their capability to infiltrate the immunosuppressive tumor microenvironment, suggest they could have a pivotal role in treating solid tumors, including CRC [313,314]. Therefore, the evolution of CAR macrophage therapy represents an exciting domain of ongoing research and future development [299,300].

In conclusion, immunotherapy’s evolution presents both opportunities and challenges for colorectal cancer treatment. The deepening understanding of the tumor microenvironment has spurred innovative therapies like CAR T-cell, NK cell, and CAR macrophage therapy. Despite hurdles like tumor antigen diversity and suppressive tumor environments, these strategies hold considerable promise. With ongoing research into tumor intricacies, we edge closer to realizing personalized treatments for solid tumors.

## 10. Culmination of CRC Tumor-Stroma Interactions in Metastasis: The Seed and Soil Hypothesis

The development of metastatic CRC significantly impacts patient prognosis and treatment outcomes, as metastatic disease is often more challenging to treat and associated with a poorer prognosis. A deeper understanding of the mechanisms underlying metastasis in CRC is essential for developing novel therapeutic strategies that can improve patient survival and their quality of life. In this context, the concept of “the seed” and “the soil” plays a crucial role in understanding the intricate interplay between cancer cells and their supportive stroma during the metastatic process.

In vessels, we can encounter Circulating Tumor Cells (CTCs). CTCs, having detached from the primary tumor, circulate within the bloodstream, contributing to the metastasis of cancer to other parts of the body. Beyond solitary CTCs, clusters of tumor cells, termed Circulating Tumor Microemboli (CTM), have also been identified in the blood. The effectiveness of the metastasis process, when involving naked tumor cells, appears to hinge on the activity of integrins. In this analogy, our “seed” is akin to a “burdock” [315]. CTMs can comprise multiple tumor cells or an amalgamation of tumor and non-tumor cells such as platelets, immune cells, or stromal cells. These clusters exhibit a higher metastatic potential compared to individual CTCs [316].

An intriguing set of cells is the hybrid cells, formed from the fusion of tumor cells and non-tumor cells. An example of these are the Cancer-Associated Macrophage-like Cells (CAMLs), created from the fusion of macrophages and cancer cells [317]. These hybrids, combining the antigen-presenting capability and motility of macrophages with the uncontrolled proliferation of cancer cells, are proposed as a potential mechanism for metastasis.

Moreover, there are Tumor-Educated Platelets (TEPs), which are not hybrids but signify a form of interaction between tumor and non-tumor cells. Tumors can emit factors that alter the RNA profile of platelets, thus ‘educating’ them. TEPs are under investigation for their potential as a cancer biomarker [318].

This perspective spawns the notion that a tumor cell begins the construction of its own stromal environment, starting from the individual cells in its vicinity. These cells are reprogrammed and potentially enveloped within a layer of host cell cytoplasm to facilitate its propagation. This can be compared to a seed, with the “seed” representing the vulnerable colon cancer cell that initially lacks a place to root. Much like seedlings, however, it has a high probability of thriving within the stroma-“soil”, an environment that shelters and nourishes the tumor cell. We hypothesize this to be among the most successful mechanisms for metastasis. By finding a method to impede the development of this “nutrient soil” around tumor cells entering the circulatory system, we could potentially counter the formidable challenge of colorectal cancer metastasis.

It is less plausible to believe that a solitary cancer cell from the original tumor could evolve into a metastatic lesion in a distant organ during metastasis. The “desert” represented by intact organs lacks the supportive stromal environment necessary for a highly immunogenic cancer cell to root and proliferate. Instead, the proposition suggests that metastasis could result from clusters of cells, consisting of cancer cells and associated stromal elements, which collectively create a suitable microenvironment for metastatic cancer cells’ survival and proliferation [219,220,221].

An example of such a metastatic unit could be the presence of CAMLs in the bloodstream. Composed of both cancer cells and stromal macrophages, CAMLs form a unique cell population. The identification of these cells in cancer patients’ blood suggests their potential role in metastasis. In this context, the cancer cell and the macrophage together constitute a primary metastatic unit capable of colonizing distant organs and establishing metastatic growths. The inclusion of the stromal macrophage in this unit could provide the necessary support and an immunosuppressive environment, enabling the highly immunogenic cancer cell to survive and proliferate within the “desert” of an intact organ (Figure 2).

The hypothesis of metastasis involving aggregates of cancer and stromal cells underlines the importance of considering both the cancer cells and their surrounding stroma when developing therapeutic strategies for metastatic CRC. By simultaneously targeting the “seed” (cancer cell) and the “soil” (supportive stromal environment), researchers could potentially interrupt the metastatic process, improving patient outcomes.

As observed, cancer cells possess a unique ability—reprogramming. This is evident in thrombocytes, which have a lifespan of 9–12 days, indicating that reprogramming can occur rapidly. A cancer cell will seize any opportunity to settle in a new location and will swiftly reprogram the surrounding cells to assume stromal-like characteristics.

Again, the significance of considering both the cancer cells and their surrounding stroma in the development of therapeutic strategies for metastatic CRC is underscored by this hypothesis of metastasis involving aggregates of cancer and stromal cells. By targeting both the “seed” and the “soil”, we can potentially disrupt the metastatic process, thereby improving patient outcomes.

(A): As a tumor progresses, a colorectal cancer (CRC) cell, triggered by mutational and epigenetic events, starts to exhibit uncontrolled proliferation and evasion from apoptosis. In the figure, the CRC cell, or the “seed”, (represented as a bright circle with a pink nucleus) thrives within the “soil” provided by cancer-associated fibroblasts (CAFs) and tumor-associated macrophages (TAMs) (1—CRC cell is surrounded and covered by pink spindle-shaped CAFs and process cells with grayish-blue cytoplasm TAM). These stromal cells create a humoral (2—depicted as “bubbles” in the picture) and cellular shield, facilitating the delivery of nutrients and exogenous growth stimuli to the tumor cells.

Key factors involved in this process include matrix metalloproteinases (MMP2, MMP9, MMP14), chemokines (CCL2, CXCL9, CXCL10), growth factors (VEGFA, FGF2, PDGFB, TGF, TNF-alpha), and cytokines (IL-1b, IL-10). These contribute to extracellular matrix remodeling, immune cell recruitment, and angiogenesis (3—as indicated by the schematic blood vessel in the figure). Surface molecules such as PD-L1, PD-1, and CTLA4 suppress T-cell activation, enable tumor cells to evade immune surveillance, and foster a supportive microenvironment for the CRC cell. In the figure, immune cells, specifically a T-cell (4) and a natural killer (NK)-cell (5), are shown. They are isolated from the CRC cell by CAFs and TAMs and are under the suppressive influence of humoral factors and macrovesicles (2—depicted as “bubbles” in the picture).

(B): This illustration highlights the interaction and invasion of Cancer-Associated Macrophage-like Cells (CAMLs) into a blood vessel (3). CAMLs, represented as a hybrid of macrophages and CRC cells (6—depicted as a mixed cell-like association of a bright CRC cell with a TAM, having a grayish-blue derivative, with an attached pink spindle-shaped CAF), utilize specific molecules such as matrix metalloproteinases (MMPs), integrins, and selectins to infiltrate the vessel wall, underlining their invasive capabilities. Furthermore, CAMLs can adhere to cancer-associated fibroblasts (CAFs) or aid tumor cells undergoing a epithelial–mesenchymal transition (EMT) to ease the invasion into the blood vessel. This visual representation showcases the intricate interactions within the tumor microenvironment that contribute to tumor progression and metastasis.

(C): The figure portrays the formation of metastasis where tumor and stromal cell associations (such as CAMLs) (6) carry the necessary “seeds” and an initial supply of “soil” to safeguard CRC cells during the early stages and prepare fertile “soil” for the protection and further proliferation of tumor cells. These cells utilize various molecular factors, such as integrins, matrix metalloproteinases (MMPs), and chemokines, to facilitate the invasion and adaptation to new tissues, promoting metastatic growth and colonization. In the figure, we see again the microenvironmental “status quo,” where immune cells, specifically a T-cell (4) and a natural killer (NK)-cell (5), are isolated from the CRC cell by the CAF and remnants of TAM in the CAML. They remain under the suppressive influence of humoral factors (2—depicted as “bubbles” in the picture). In the figure, fibroblasts (7—depicted as pink spindle cells) and a macrophage (8—depicted as process cells with grayish-blue cytoplasm) represent the future origins of CAFs and a TAM, respectively.

## 11. Conclusions

The conception of a tumor as a unified structure, constituted of intricately woven tumor cells that strive for heterogeneity, a diverse population of stromal cells, the extracellular matrix space, and subjected to external influences such as chemotherapy and immunotherapy strategies, offers a profound perspective into the interaction of various elements contributing to colorectal cancer development and progression.

This complex panorama brings forth several promising directions for future scientific exploration. One such trajectory involves the study of the precise interactions within the triad of the microbiome, gut-associated lymphoid tissue, and colorectal cancer using cutting-edge ‘omics’ methodologies. This line of investigation could offer considerable insights into the roles these factors embody in the initiation and progression of colorectal cancer, thereby identifying potential targets for future therapeutic approaches.

Alterations in the Wnt signaling pathway, the phenomena of tumor budding, and the existence of Conserved Oncogenic Signatures in the stroma of colorectal cancer underscore the necessity of understanding the genetic and cellular mechanisms propelling the progression of colorectal cancer. This comprehension is crucial for the development of targeted therapies. For example, a comprehensive exploration into the role of Wnt signaling in tumor budding, which is acknowledged as a potential prognostic factor, may unravel targets for therapies aimed at reducing colorectal cancer progression and metastasis. Similarly, the employment of machine learning algorithms for the identification of Conserved Oncogenic Signatures may lay the groundwork for the creation of personalized treatment plans.

Significant challenges in colorectal cancer treatment include the severe conditions of hypoxia and the incidence of multidrug resistance. The development of therapies targeted at hypoxia, which have the capacity to modify tumor responses to hypoxia or to exploit the metabolic vulnerabilities of hypoxic cancer cells, could aid in circumventing treatment resistance. The exploration of cutting-edge biotechnological strategies, such as systems of nanocarriers for drug delivery, could offer effective solutions to bypass mechanisms of multidrug resistance.

The ‘Seed and Soil’ hypothesis offers a useful viewpoint on the metastatic process, highlighting the need to disrupt both the cancer cells (referred to as the ‘seeds’) and their supportive stromal environment (referred to as the ‘soil’). A thorough investigation into the molecular mechanisms and signaling pathways governing these interactions could potentially disrupt the metastatic process.

Immunotherapies, comprising checkpoint inhibitors, Chimeric Antigen Receptor T-cell therapy, and Natural Killer cell therapy, offer promising new paths for colorectal cancer treatment. As our understanding of the tumor microenvironment and immune responses continues to broaden, so does the potential for developing personalized treatment strategies. This could lead to substantial improvements in outcomes for colorectal cancer and other solid tumors. Future research could focus on transforming the stroma of colorectal cancer from an immune-exclusionary state to an immune-infiltrated state, potentially opening up new opportunities for Stromal Immunotherapy.

In conclusion, the management of colorectal cancer necessitates a multifaceted approach that incorporates an understanding of and strategies for targeting the complex interplay of microbiome dysbiosis, immune responses, genetic aberrations, cellular mechanisms, and the tumor microenvironment. The continuing evolution and integration of immunotherapies provide renewed hope for patients with colorectal cancer. As we continue to untangle the complexities of colorectal cancer, it becomes increasingly evident that a comprehensive approach integrating these targeted strategies is essential for effective disease management.

Despite the promising advancements made thus far, further research is needed to fully realize the potential of these therapeutic strategies. An integrative and comprehensive approach will be key to significantly improving the prognosis and treatment outcomes of colorectal cancer. By expanding the immunotherapeutic arsenal beyond immune checkpoint inhibitors and Chimeric Antigen Receptor T-cell therapies to include other forms of immunotherapy, such as tumor vaccines or oncolytic viruses, and by increasing our understanding of resistance mechanisms to current immunotherapies, we could potentially facilitate the development of combination therapies that enhance their effectiveness.

## Figures and Tables

**Figure 1 biomedicines-11-02361-f001:**
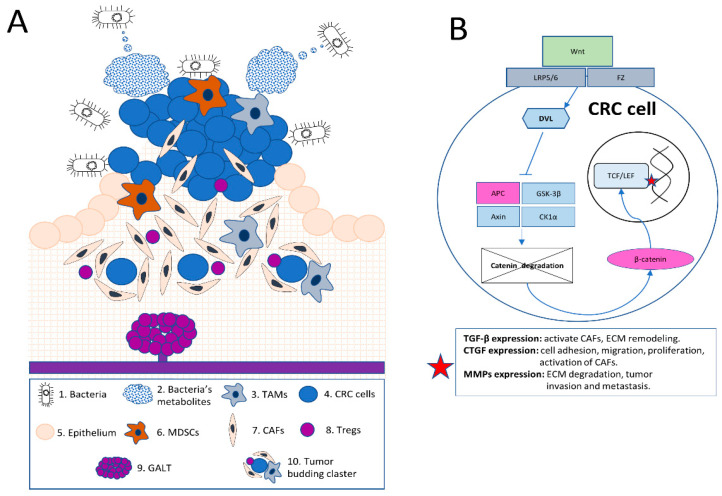
Unique Features and Complexity of the Colorectal Tumor Microenvironment (TME).

**Figure 2 biomedicines-11-02361-f002:**
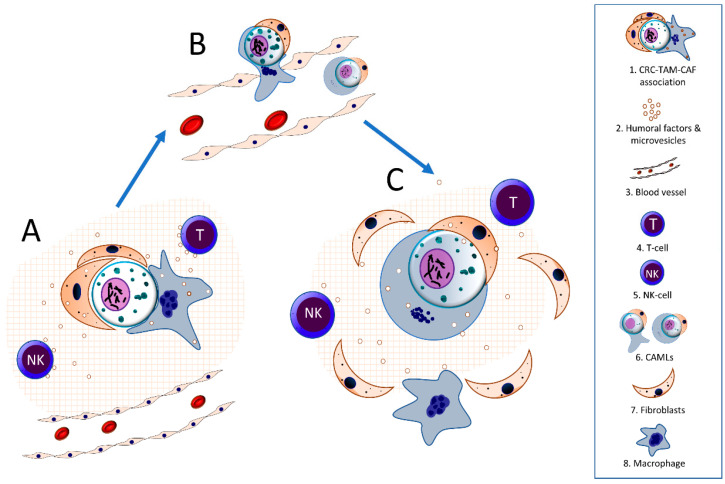
CRC Metastasis: Tumor-Stroma Interactions and the Seed and Soil Hypothesis.

**Table 1 biomedicines-11-02361-t001:** Bacteria of gut microbiome with pro- and anticancer activities.

Bacteria	Mechanism of Action	References
**Preliminary Pro-Cancer**		
*Fusobacterium nucleatum*	Induces DNA damage and genetic changes, promotes cytokine production, influences immune regulation, possibly enhancing CRC progression.	[23,24,31,32]
*Bacteroides fragilis*	Produces cytotoxic BFT, alters cellular structures, induces inflammation and activates signaling pathways, triggers changes in host defense mechanisms contributing to CRC.	[33,34]
*Enterococcus faecalis*	Utilizes biliverdin to promote CRC cell proliferation and angiogenesis, induces immunomodulation, causes genomic instability and disrupts intestinal barrier, contributing to CRC progression.	[35,36,37]
*Escherichia coli* (phylotype B2, genotoxic pks + *E. coli*)	Overrepresented cytotoxic phenotype contributes to DNA damage, promotes carcinogenic effects via the production of colibactin mediated by the pks gene.	[38,39,40]
*Peptostreptococcus anaerobius*	Adheres to CRC cells via PCWBR2 protein, activates PI3K–Akt–FAK pathway, promotes cell proliferation and triggers pro-inflammatory responses, enhancing CRC progression.	[41,42]
*Streptococcus gallolyticus*	Adheres to host cells via Type VII secretion system, stimulates cell proliferation and promotes CRC via upregulation of β-catenin, c-Myc, and PCNA.	[43,44,45]
*Clostridium septicum*	Exacerbates CRC through α-toxin production, induces necrosis and mucosal ulceration, impairs immune response, fostering a conducive environment for CRC.	[46,47,48]
**Preliminary Anti-Cancer**		
*Ruminococcus gnavus*	Reduces tumor growth and degrades inhibitory compounds like lyso-glycerophospholipids, enhancing the activity of CD8+ T cells, potentially mitigating CRC progression.	[49,50]
*Bifidobacterium longum*	Modulates oncogenic and tumor suppressor miRNAs, suppresses pro-inflammatory cytokines, enhances adhesion to the intestinal tract, increases short-chain fatty acids production, and improves intestinal barrier function, potentially mitigating CRC progression.	[51,52,53,54]
*Lactobacillus acidophilus*	Induces apoptosis in CRC cells, mitigates ulcerative colitis via increased acetate production and control of inflammation, potentially reducing CRC progression.	[51,52,55,56,57]
*Lactobacillus rhamnosus*	Triggers apoptosis in cancer cells, boosts immune responses, increases carcinoembryonic antigen secretion from cancer cells, modulates gut immune landscape by increasing CD8 T-cell responses, potentially mitigating CRC progression.	[52,58,59,60,61]
*Faecalibacterium prausnitzii*	Reduces formation of aberrant crypt foci, suppresses lipid peroxidation levels, inhibits CRC cell proliferation, enhances gut microbiota diversity, produces butyrate to augment tumor-suppressing effects, potentially mitigating CRC progression.	[62,63]
*Bifidobacterium breve*	Stimulates immune response by increasing cytotoxic CD8+ T cells, promotes production of anti-tumor cytokines, potentially reducing CRC progression.	[49,64,65,66]
*Lactobacillus reuteri*	Provokes caspase-9-dependent apoptosis in tumor cells, inhibits cell invasion and proliferation, reduces proliferation and survival in colon cancer cells with its metabolite, reuterin, potentially mitigating CRC progression.	[52,67,68,69]
*Bifidobacterium adolescentis*	Suppresses colorectal carcinogenesis, inhibits harmful bacterial enzymes such as β-glucuronidase, β-glucosidase, tryptophanase, and urease, differentially regulates Treg/Th17 immune responses, potentially reducing CRC progression.	[52,70,71,72]
*Lactobacillus plantarum*	Strengthens the intestinal mucosal barrier by regulating occludin and claudin-1 proteins, inhibits harmful bacterial enzymatic activity, regulates CRC cell proliferation and apoptosis, potentially mitigating CRC progression.	[52,73,74,75]

Note: The presented table effectively illustrates the dual role of gut microbiota in the progression and mitigation of colorectal cancer (CRC). The ‘Preliminary Pro-cancer’ section identifies bacteria such as Fusobacterium nucleatum, Bacteroides fragilis, among others, that may contribute to CRC progression through various mechanisms including DNA damage, inflammation, and immune response impairment. On the other hand, the ‘Preliminary Anti-cancer’ section lists beneficial bacteria, including multiple Bifidobacterium and Lactobacillus species, which exhibit potential anti-carcinogenic activities, thereby possibly mitigating CRC progression. It is important to note, however, that the roles of these bacteria are not exclusive to CRC. They have multifaceted implications across a broad spectrum of health and disease states beyond CRC, underlining the complex and integral relationship between the gut microbiome and overall human health.

**Table 2 biomedicines-11-02361-t002:** Cytokines and Growth Factors Involved in CRC Stroma.

Gene	Full Name	Role in CRC Stroma	References
*MMP2*	Matrix Metalloproteinase 2	ECM remodeling, degrades various ECM components, facilitates tumor cell invasion and metastasis.	[200,201]
*MMP9*	Matrix Metalloproteinase 9	ECM remodeling, degrades collagen and other ECM components, promotes tumor cell invasion, supports angiogenesis.	[200,201]
*MMP14*	Matrix Metalloproteinase 14	ECM remodeling, involved in the cleavage of cell surface proteins and the breakdown of ECM components, promotes tumor invasion and angiogenesis.	[169,202]
*LOX*	Lysyl Oxidase	ECM remodeling, catalyzes the cross-linking of collagens and elastin, contributes to the stiffening of the tumor microenvironment and promotes tumor progression.	[203,204]
*FN1*	Fibronectin	ECM remodeling, involved in cell adhesion, migration, and proliferation; its increased expression is associated with tumor progression and poor prognosis in CRC.	[205]
*COL1A1*	Collagen Type I Alpha 1 Chain	ECM remodeling, major structural component of the ECM, its increased expression is associated with tumor progression and poor prognosis in CRC.	[206]
*COL3A1*	Collagen Type III Alpha 1 Chain	ECM remodeling, another structural component of the ECM, its increased expression is associated with tumor progression and poor prognosis in CRC.	[207]
*COL5A1*	Collagen Type V Alpha 1 Chain	ECM remodeling, another structural component of the ECM, its increased expression is associated with tumor progression and poor prognosis in CRC.	[208]
*PD-L1*	Programmed Death-Ligand 1	Immune checkpoint molecule, inhibits T cell activation, promotes immune evasion by tumor cells.	[209,210,211]
*PD-1*	Programmed Cell Death Protein 1	Immune checkpoint receptor, dampens immune response, allows tumor cells to escape immune surveillance.	[209,210,211]
*CTLA4*	Cytotoxic T-Lymphocyte-Associated Protein 4	Immune checkpoint receptor, inhibits T cell activation, contributes to immune evasion by tumor cells.	[212,213]
*CXCL9*	Chemokine (C-X-C motif) Ligand 9	Recruits immune cells, such as T cells and natural killer cells, to the tumor microenvironment; enhanced anti-tumor immunity.	[214]
*CXCL10*	Chemokine (C-X-C motif) Ligand 10	Recruits immune cells, such as T cells and natural killer cells, to the tumor microenvironment; enhanced anti-tumor immunity.	[214,215]
*CCL2*	Chemokine (C-C motif) Ligand 2	Recruitment of monocytes, macrophages, and other immune cells to the tumor site; altered expression associated with immune cell infiltration and tumor progression.	[216,217]
*IFNG*	Interferon Gamma	Activates and modulates immune response against tumor cells, affects expression of immune checkpoint molecules and other immune-related genes.	[218]
*VEGFA*	Vascular Endothelial Growth Factor A	Promotes growth of new blood vessels from existing vasculature, stimulates endothelial cell proliferation, migration, and survival.	[219,220,221]
*VEGFR2*	Vascular Endothelial Growth Factor Receptor 2 (KDR)	Primary receptor for VEGFA on endothelial cells, activation by VEGFA leads to a signaling cascade promoting angiogenesis and vascular permeability.	[222,223]
*ANGPT1*	Angiopoietin-1	Regulates angiogenesis by binding to the endothelial cell receptor tyrosine kinase, Tie2, promotes vessel maturation and stability.	[224,225]
*ANGPT2*	Angiopoietin-2	Acts as an antagonist of ANGPT1, binds to Tie2, promotes vessel destabilization and sprouting angiogenesis.	[225,226]
*FGF2*	Fibroblast Growth Factor 2 (bFGF)	Regulates angiogenesis, stimulates endothelial cell proliferation, migration, and differentiation, acts synergistically with VEGFA to promote blood vessel formation.	[227,228]
*PDGFB*	Platelet-Derived Growth Factor B	Promotes recruitment of pericytes to newly formed blood vessels, essential for blood vessel maturation and stabilization.	[229,230]

## Data Availability

All data and materials are available upon reasonable request. Address to I.B. (email: buchwalow@pathologie-hh.de) or M.T. (email: mtiemann@hp-hamburg.de), Institute for Hematopathology, Hamburg, Germany.

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
