# Peer review of "Cellular and Molecular Mechanisms of the Tumor Stroma in Colorectal Cancer: Insights into Disease Progression and Therapeutic Targets"

_biomedicines, 2023, doi:10.3390/biomedicines11092361_

Round 1
Reviewer 1 Report
In this manuscript, Shakhpazyan et al. have reviewed the role of stromal components in colorectal cancer. They have addressed the review from a clinical and translational perspective. This is an interesting manuscript, although there are several areas that need to be improved. Here are my observations:
- The authors state in the abstract that this is a systematic review. However, they need to expand on and explain the criteria used for this systematic review, as mentioned in the PRISMA guidelines.
- The authors need to improve the manuscript's flow and structure. They should combine smaller paragraphs into larger ones and present a current picture of each subsection.
- The authors need to add appropriate references to the sentences, specifically on page 2.
- The authors need to add new columns to Table 1 and mention the functions of each bacteria, or any other biological role relevant to colorectal cancer, with references.
- The authors need to completely restructure the conclusion section. The conclusion should be the crux of the entire article and should provide future directions. Therefore, the expansion of CAR T cells and NK cells should ideally be shifted to a new section in the separate review subsection.
- The authors need to add a separate subsection on immunotherapy (checkpoint blockade).
The authors need to improve the overall structure of the manuscript and merge smaller paragraphs into coherent structures.
Author Response
Dear Reviewer,
We sincerely appreciate your feedback and advice regarding our manuscript. Following, we provide point-by-point responses to your comments:
- The authors state in the abstract that this is a systematic review. However, they need to expand on and explain the criteria used for this systematic review, as mentioned in the PRISMA guidelines.
Initially, we did not intend to position our review as a systematic review. To avoid any misunderstandings, we have removed any mentions of 'systematic' or similar words from the abstract.
- The authors need to improve the manuscript's flow and structure. They should combine smaller paragraphs into larger ones and present a current picture of each subsection
We have endeavored to improve the text structure by eliminating the highlighted minor sub-points within the chapters.
- The authors need to add appropriate references to the sentences, specifically on page 2.
We have enriched our manuscript with additional references.
- The authors need to add new columns to Table 1 and mention the functions of each bacteria, or any other biological role relevant to colorectal cancer, with references.
Table 1 has been revised with a focus on the mechanisms of pro-cancerous and anti-cancerous effects of bacteria.
- The authors need to completely restructure the conclusion section. The conclusion should be the crux of the entire article and should provide future directions. Therefore, the expansion of CAR T cells and NK cells should ideally be shifted to a new section in the separate review subsection.
We have restructured the conclusion, and aspects related to immunotherapy have been moved to Chapter 9.
- The authors need to add a separate subsection on immunotherapy (checkpoint blockade)
We have created a separate Chapter 9, dedicated to aspects of the stromal influence on immunotherapy, including checkpoint blockade.
Reviewer 2 Report
Comments:
1. Please provide impact of tumor-stroma ratio on CRC?
2. List a Table of good vs bad bacteria associated with CRC. Is Table 1 specific for CRC?
3. Any updated p53 and Bcl-2 on tumor budding in CRC?
4. Any new WT and mutants of p53-CRC info in recent years?
Author Response
Dear Reviewer,
We sincerely appreciate your feedback and advice regarding our manuscript. Following, we provide point-by-point responses to your comments:
- Please provide impact of tumor-stroma ratio on CRC.
We have reviewed the topic of the potential significance of the tumor-stroma ratio in Chapter 1, underscoring the comprehensive importance of this parameter, which includes the stroma as a whole, and the contrasting findings from various researchers regarding the prognostic value of the tumor-stroma ratio.
- List a Table of good vs bad bacteria associated with CRC. Is Table 1 specific for CRC?
The microbiota presented in Table 1 is not specific to colorectal cancer. It also affects other processes, for example, the inflammatory response in the intestine, shifting the balance towards favoring or inhibiting carcinogenesis. An explanation on this topic has been added to the manuscript in the form of a footer to the table.
- Any updated p53 and Bcl-2 on tumor budding in CRC?
We have incorporated up-to-date data regarding the impact of p53 and Bcl-2 statuses on tumor budding in the relevant chapter - Chapter 5.
- Any new WT and mutants of p53-CRC info in recent years?
We have explored and discussed how the mutation status of the TP53 gene (which codes for the tumor suppressor p53) affects the interaction between tumor cells and the stroma. We found that TP53 mutations influence tumor development through highly intricate and fascinating mechanisms. Considering that tumor cells carry the somatic TP53 mutations, and also in light of the discussion on the statuses of p53 and Bcl-2 in this chapter, we decided to delve deeper into this topic in the chapter on budding (Chapter 5), having sourced relevant literature on the subject.
Round 2
Reviewer 1 Report
The authors have improved the manuscript.
The authors need to revise the manuscript to improve minor grammatical inconsistencies.
Reviewer 2 Report
No more comments